# Deformable Photonic Crystals Based on Chiral Liquid Crystals with Thermal-Mediative Shape Memory Effect

**DOI:** 10.3390/ma16010035

**Published:** 2022-12-21

**Authors:** Min-Seok Park, Kitae Kim, Young-Joo Lee, Jun-Hee Na, Se-Um Kim

**Affiliations:** 1Department of Electrical and Information Engineering, Seoul National University of Science and Technology, 232 Gongneung-ro, Nowon-gu, Seoul 01811, Republic of Korea; 2Department of Convergence System Engineering, Chungnam National University, Daejeon 34134, Republic of Korea; 3Department of Materials Science and Engineering, University of Pennsylvania, 3231 Walnut Street, Philadelphia, PA 19104, USA

**Keywords:** chiral liquid crystals, shape memory effect, photonic crystals, reactive mesogens

## Abstract

We propose a deformable photonic crystal that exhibits the thermal-mediative shape memory effect. The chiral liquid crystalline polymeric scaffold, which produces the structural colors from a helical twist of the liquid crystal director, is prepared through phase-stabilization of a reactive mesogen in a small molecular chiral liquid crystal (CLC), polymerization, and removal of the CLC. The prepolymer of polyurethane acrylate (PUA) is then infiltrated in the prepared scaffold and subsequently photo-polymerized to form a CLC-PUA composite film. Upon compression, this film shows the blue shift of the structural color and retains this color-shift as released from compression. As the temperature increases, the color is recovered to a pristine state. The concept proposed in this study will be useful for designing mechanochromic soft materials.

## 1. Introduction

Photonic crystals (PCs) show high-intensity Bragg colors by the photonic band gap effect produced by the periodic refractive index [1,2,3]. The coloration capability of PCs allows a straightforward perception of the degree of the deformation since the central wavelength of the Bragg color is associated with the periodicity and/or the value of the refractive index. The recovery of the deformed PCs to the pristine state for reuse, however, often requires a continuous application of stimuli input or the change in the environment surrounding the PCs. Also, the increase of the elasticity of PCs for reversibility may not allow the storage of the color shift at the deformed state or may decrease the contrast and the magnitude of refractive index oscillation due to material limitations.

PCs with shape memory effects can achieve remote reversibility on demand without a decrease in coloration performance. Shape memory polymers (SMPs) can recover their deformation to the retained shape by external stimuli such as temperature [4,5,6], electric or magnetic field [7,8], light [9], solvent [10,11], or mechanical force [12]. SMPs of such stimuli-responsive transformation, when accompanied by their softness, lightweight, and bio-compatibility, can be used in sensors [13,14,15], actuators [16,17], wearables [6,16,18], or bio-medical applications [19,20]. To realize shape memory photonic crystals (SMPCs), SMPs can be structured in an inverse opal structure, that is the three-dimensionally ordered pores surrounded by the SMP frame [12,21]. This structure is suitable for achieving shape memory effects because materials without shape memory effects are not used. However, the construction of spatially homogeneous inverse opal structures has been particularly challenging due to the time-consuming and high-cost patterning processes.

In contrast to trivial PCs, chiral liquid crystals (CLCs) generate structural colors from the helical rotation of anisotropic liquid crystal (LC) molecules [22,23]. The helical pitch (*P*) denotes the periodicity of this helical rotation. In CLCs, the central wavelength of the structural color (*λ*) is given as *λ* = (*n*_o_ + *n*_e_)*P*/2, where *n*_o_ and *n*_e_ are the ordinary and the extraordinary refractive indices of the LC molecules, respectively. The CLC polymers can act as a color-generating template, and other materials are injected into those CLC polymers to provide the stimuli-responsive color-changing functions upon the electric field, humidity, or temperature. However, the shape memory effect has not been achieved yet so far in such composite systems.

Here, we propose an SMPC based on a CLC. We first fabricate a CLC template and then inject a polyurethane acrylate (PUA) prepolymer to comprise a CLC-PUA composite film. The PUA filling and the CLC template play a separate role in realizing the thermal-mediative shape memory effect and the structural color, respectively. Therefore, the CLC-PUA composite film exhibits both functions simultaneously without any prevalent patterning processes. The CLC template is naturally rigid, while the CLC-PUA composite film becomes deformable owing to the increased elasticity from the PUA filling. The structural color of the CLC-PUA composite film changes upon deformation due to the variation in the helical pitch, and it can be recovered to the color in the pristine state by simple heating. Also, the CLC-PUA composite film can display the spatially patterned compression by the different magnitudes of color shift.

## 2. Materials and Methods

### 2.1. Materials

Figure 1 shows the chemical structure of liquid crystalline materials and components of the PUA prepolymer. 2,2-dimethoxy-2-phenyl acetophenone (DMPA) and poly(vinyl alcohol) (Mw 31,000~50,000, 98~99% hydrolyzed) (PVA) were purchased from Sigma Aldrich. Dichloromethane (DCM) and ethanol were purchased from Samchun Chemicals Co., Ltd. 2-Methyl-1,4-phenylene bis(4-(3-(acryloyloxy)propoxy)benzoate) (RM257) and 4-cyano-4-pentylbiphenyl (5CB) were purchased from Henan Wentao Chemical Product Co., Ltd. (R)-2-Octyl 4-[4-(Hexyloxy)benzoyloxy]benzoate (R811) and (S)-2-Octyl 4-[4-(Hexyloxy)benzoyloxy]benzoate (S811) were purchased from Nanjing Aocheng Chemical Co., Ltd., Nanjing, China. The PUA prepolymer, MINS-301RM, was purchased from Minuta Tech. It consists of hydroxyethyl methacrylate (HEMA), 1,6-hexanediyl ester (HDDA), trimethylolpropane triacrylate (TMPTA), urethane acrylate oligomer, and 2-hydroxy-2-methyl-1-phenyl-1-propanone.

### 2.2. Methodes

#### 2.2.1. Preparation of the CLC-PUA Composite Film

Figure 2 shows the fabrication process of the CLC-PUA composite film. Glass substrates for preparing the LC cell were cleaned by sonication in acetone for 20 min, rinsing with distilled water, drying, and undergoing UV ozone treatment (Harrick Expanded Plasma Cleaner & PlasmaFlo™) for 5 min. A solution of 1 wt.% PVA in distilled water was prepared for a homogeneous alignment layer. This solution was spin-coated on glass substrates at the rate of 2000 rpm for 30 s, followed by annealing at 100 °C for 30 min. The PVA films deposited on top and bottom glass substrates were rubbed in an anti-parallel direction. The LC cell was constructed using those top and bottom glass substrates with 20 μm or 100 μm cell gap.

The CLC prepolymer comprising 5CB, RM257, and R811 (right-handed; RH chiral dopant) or S811 (left-handed; LH chiral dopant), with a 1 wt.% doping of DMPA was dissolved in 2 mL DCM to make homogeneous phase. The concentration of the chiral dopant was varied in several experimental cases to modify the central wavelength of the structural color. The concentration of RM257 was fixed as 30 wt.% to prepare mechanically stable CLC templates. DCM was then evaporated upon stirring at room temperature for 2 h. The CLC prepolymer was injected into the LC cell by capillary action and polymerized using 365 nm UV light (Ushio Shenzhen, Inc., Shenzhen, China) with the intensity of 10 mW/cm^2^ for 10 min. The LC cell was disassembled. The CLC template was selectively formed on the top substrates that faced UV light. The residual unreactive materials after polymerization, which are 5CB and R811, were removed by immersing the CLC template in ethanol for 1 h. The CLC template was fully dried at room temperature.

A few drops of the PUA prepolymer were placed on the surface of the CLC template and infiltrated into the template by capillary effect. Another glass substrate was placed on top of the PUA prepolymer drops for the planarization. The PUA prepolymer was then polymerized using 365 nm UV light (Ushio Shenzhen, Inc.) with an intensity of 10 mW/cm^2^ for 10 min. Finally, the planarization glass was manually disassembled.

#### 2.2.2. Characterization

We examined the optical characteristics of the CLC prepolymer, the CLC template before and after the removal of residual unreactive materials, and the CLC-PUA composite films. A customized UV-vis. spectroscope (Ocean Optics) was used for measuring transmittance spectra. A polarized optical microscope (POM) (Leica, DM750P) was used for capturing the surface morphology and colors of the CLC-PUA composite film.

## 3. Results and Discussion

Figure 3 shows the transmittance spectra at Step 2 (after injection of the CLC prepolymer) and Step 3 (after photo-polymerization of the CLC prepolymer) in LC cells with a 20 μm cell gap. Before the polymerization process, the transmittance spectrum shows a photonic bandgap at the wavelength range of 850~950 nm for the CLC prepolymer with a 7 wt.% concentration of the chiral dopant. We note that the photonic bandgap of CLCs shows theoretically 50% transmittance (or 50% reflectance) instead of 0% transmittance (or 100% reflectance), because they selectively reflect the circular polarization component with the handedness identical to that of the helical structure of CLCs. The spectrum in Step 2 shows a wider photonic bandgap when compared to small molecular CLCs [24]. It results from the non-homogeneous dispersion of the reactive mesogen in the CLC prepolymer. During the polymerization, the reactive mesogen is consumed faster at the top surface region facing the UV light than at the bottom region. It leads to the dense concentration of CLC polymer on the top surface region, accompanying a blue shift of the central wavelength of the photonic bandgap [25]. Both the CLC prepolymers with RH and LH helical structures show a consistent magnitude of the wavelength shift after polymerization.

Figure 4 shows the sample images at Step 5 (injection of the PUA prepolymer, planarization, and photo-polymerization) and respective transmittance spectra. Both RH and LH CLC templates were prepared using the CLC prepolymer of an increased concentration of the chiral dopant (from 7 to 12 wt.%) to produce the structural color in the visible range. The cell gap for those films was 20 μm. The PUA prepolymer was injected into a LH CLC film (left inset) and a stack of the RH and LH CLC templates (right inset). Since the PUA prepolymer is optically isotropic, the injection resulted in a decrease in the peak intensity of the photonic bandgap. The helical structure of different handedness reflects two orthogonal circular polarizations, such that the stack of RH and LH CLC templates achieves polarization-independent reflection with an increased peak intensity. The decrease of the transmittance at the range other than the photonic bandgap wavelength primarily resulted from the index mismatch at the intermediate layer between the RH and LH CLC templates (see the schematic description of the structure in the right inset). The decrease in the transmittance can be circumvented by fabricating both layers as a homo-planar structure or as a continuous structure in a single LC cell [26]. The CLC template formed by the homo-polymerization of high-density acrylate is naturally glassy and is therefore highly brittle when deformed. Comprising the composite structure by injecting PUA filling contributes to increasing the elasticity of the film. Therefore, the magnitude of the color shift becomes higher by the increased elasticity.

Figure 5 shows the thermal-mediative shape memory effect of the CLC-PUA composite film. We compressed it using a cylindrical-shape (Figure 5a) and a spherical-shape compression applier (Figure 5b). In the case of the spherical-shape compression applier, the protrusions of different heights were patterned on the surface (inset of Figure 5c). We examined the change in the structural colors of the CLC-PUA composite film and the shape memory effect upon heating. Here, we used a 100-μm-thick CLC-PUA composite film to prevent any fracture on the film. This thickness value is too high to make a uniform helical structure. Instead, the CLC-PUA composite film scatters the structured color by focal conic state [27]. Therefore, the CLC-PUA composite film has almost no transmission at the pristine state. However, when compressed, the helical pitch is mechanically aligned to make the structural color (Figure 5a,c). Upon heating, the CLC-PUA composite film recovered to the unpressed state so that it loses its structural colors. When applying the compression with patterned protrusions of different heights, the resultant compressions at different regions are well distinguished. The resolution in such spatially varying compression or any different types of strain can be enhanced by increasing the softness of the film or making mechanically isolated pixels. In Figure 5, we measured the average full recovery time as 7 s at the threshold temperature of 80 °C, which is comparable to previous SMPCs [28].

Figure 6 shows the recovery of the fracture of the CLC-PUA composite film. The mechanical fracture was formed by applying the compression using a spherical-shaped applier without any protrusions. Here, we used a 20-μm-thick CLC-PUA composite film. In contrast to the thicker film shown in Figure 5, this film shows a clear structural color at the pristine state (yellow). The color of the compressed region, which is shown as the black dashed circle in Figure 6, is shifted from yellow to green due to the decrease in the helical pitch of the CLC polymer. As the temperature increases to 80 °C, the fracture is closed, and the color of the compressed region recovers to yellow. The crack recovery ratio can be defined as (*w*_o_ − *w*_c_)/*w*_o_, where *w*_o_ and *w*_c_ are the width of the opened crack and the closed crack, respectively [29]. The crack recovery ratio is measured to be 93%, which may be enhanced when the elasticity is further increased by using LC elastomers [23]. We observed no distortion or subsequent change of the recovered state after the temperature decreased. The recovery time in this 20 μm thick film shows no significant variation from that in the 100 μm thick film (~7 s). We note that the remaining scratch can be removed by functionalizing the prepolymers with self-healable groups [30].

## 4. Conclusions

We proposed a new concept of SMPC based on using the composite of CLC and PUA. Unlike the previous SMPCs that use the complex and non-uniform fabrication approach, our SMPC shows scalable yet highly uniform colors from the self-aligning capability of the CLC. The elasticity and the thermal-mediative shape memory effect are separately introduced by injecting PUA prepolymers into the CLC template. In turn, our SMPC shows the strain-responsive color change and its remote recovery without using any opposite strain but the simple heating process. Our approach of combining two different materials working independently to make stimuli-responsive color and shape memory effect provides much room for customizing both the optical property and the mechanical property. The concept proposed here will play an important role to develop mechanochromic displays and sensors.

## Figures and Tables

**Figure 1 materials-16-00035-f001:**
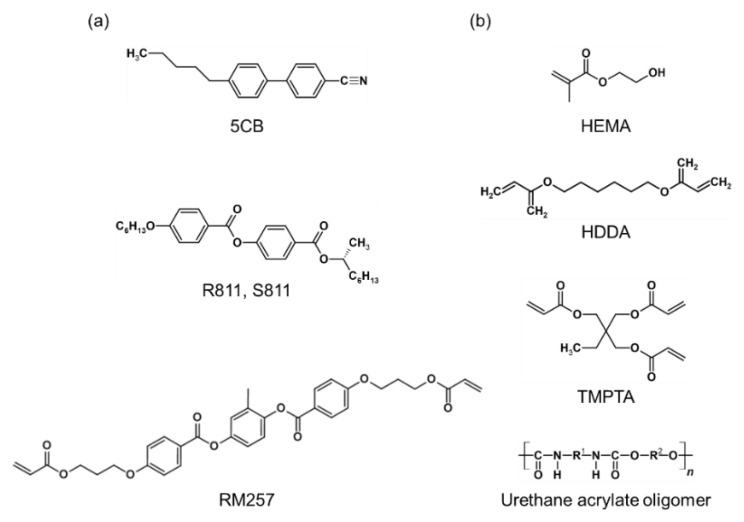
Chemical structures of materials used for fabricating the CLC-PUA composite film. (**a**) The chemical structures of liquid crystalline materials; the small molecular LC (5CB), the chiral dopant (R811 and S811), and the reactive mesogen (RM257). (**b**) The chemical structures of representative components of the PUA prepolymer: HEMA, HDDA, TMPTA, and urethane acrylate oligomer.

**Figure 2 materials-16-00035-f002:**
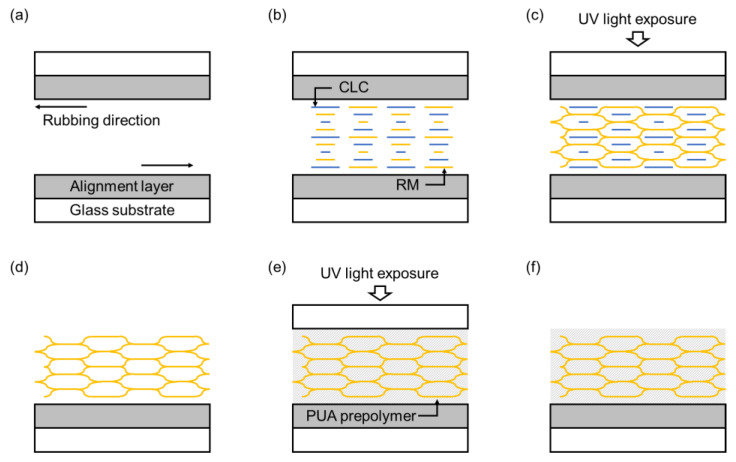
Schematic diagram showing the fabrication process of the CLC-PUA composite film. (**a**) Step 1: Assembly of the LC cell with anti-parallel rubbing. (**b**) Step 2: Injection of the CLC prepolymer. (**c**) Step 3: Photo-polymerization of the CLC prepolymer. (**d**) Step 4: Removal of residual unreactive materials. (**e**) Step 5: Injection of the PUA prepolymer, planarization, and photo-polymerization. (**f**) Step 6: Removal of the planarization glass.

**Figure 3 materials-16-00035-f003:**
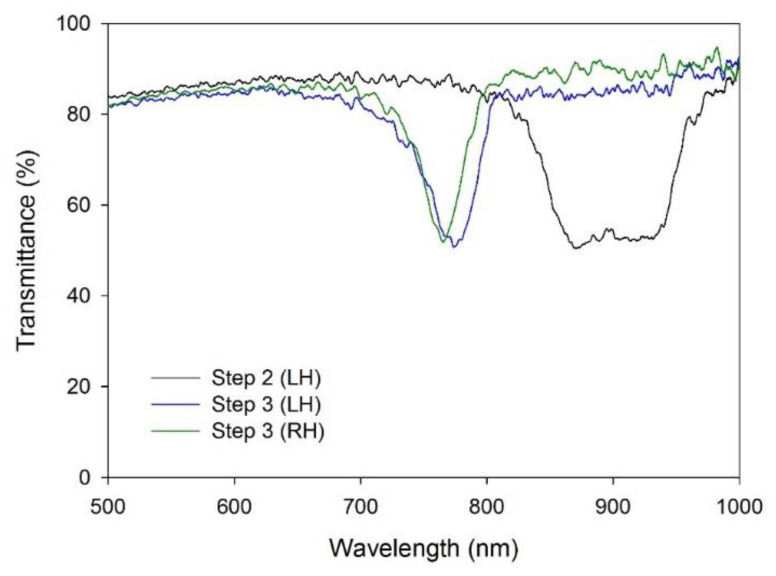
Transmittance spectra of the CLC prepolymer at Step 2 (after injection of the CLC prepolymer) (black line) and the RH (green line) and LH (blue line) CLC templates at Step 3 (after photo-polymerization of the CLC prepolymer).

**Figure 4 materials-16-00035-f004:**
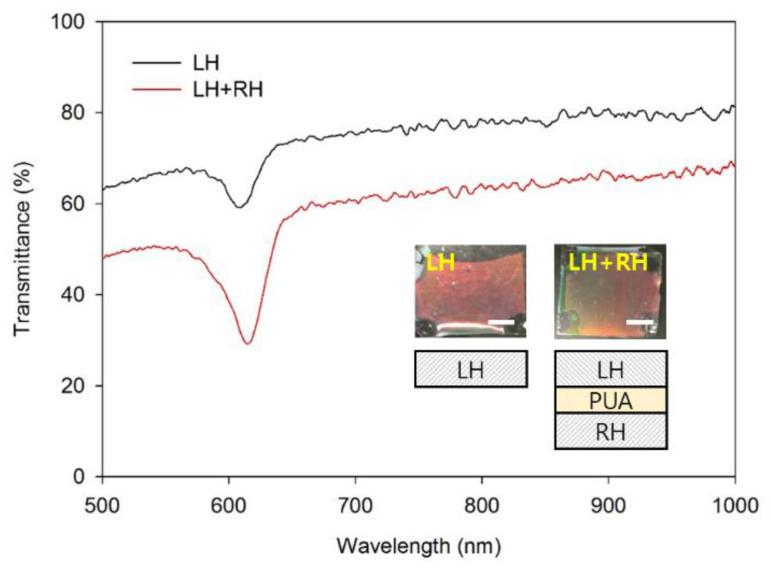
The optical characteristics of the CLC-PUA composite films. Transmittance spectra of CLC-PUA composite films with a LH CLC template (black line) and a stack of LH and RH CLC templates (red line). Insets show the optical image of the CLC-PUA composite film with a LH CLC template (left) and a stack of LH and RH CLC templates (right) with their schematic descriptions. Scale bares are 200 μm.

**Figure 5 materials-16-00035-f005:**
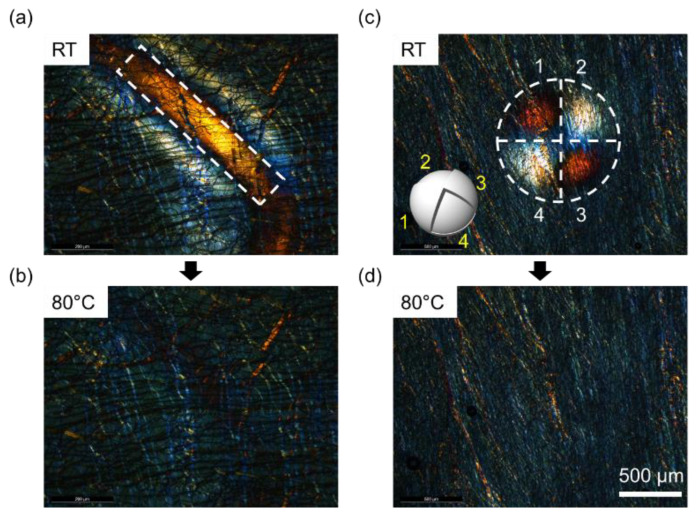
The thermal-mediative shape memory effect of the CLC-PUA composite film (100 μm thick). (**a**,**b**) POM images of the CLC-PUA composite film (**a**) upon the compression using a cylindrical-shape compression applier, and (**b**) recovered to the pristine state by heating. (**c**,**d**) POM images of the CLC-PUA composite film (**c**) upon the compression using a spherical-shape compression applier with a patterned protrusion (inset), and (**d**) recovered to the pristine state by heating. In (**c**), the numbers represent the labeled protrusions and the compressed regions by corresponding protrusions.

**Figure 6 materials-16-00035-f006:**
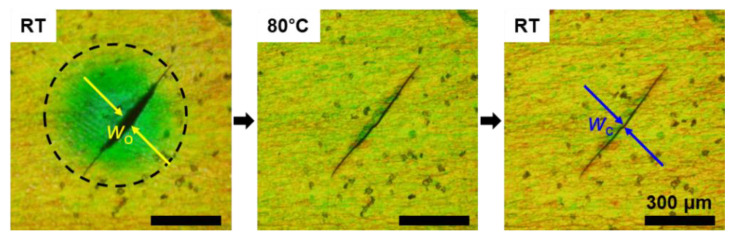
The reflection mode POM images showing the mechanical fracture recovery of the CLC-PUA composite film (20 μm thick) by the thermal-mediative shape memory effect. A fracture was formed by the compression (black dashed circle) at room temperature, recovered back to the initial state at 80 °C, and remained unchanged from the recovered state as the temperature decreased to room temperature.

## Data Availability

Not applicable.

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
