# Peer review of "Deformable Photonic Crystals Based on Chiral Liquid Crystals with Thermal-Mediative Shape Memory Effect"

_materials, 2022, doi:10.3390/ma16010035_

Round 1
Reviewer 1 Report
The manuscript on memory effect using LC materials is a nice and upgrowing field. The authors doing good work. Although the paper, seem to me, is written within short period of time. My comments, which need to be address before publishing the manuscript in the journal, are as follows:
1. The results and observation are quite good but scientific soundness should be improved via more discussion about , specifically about the Figure 4, 5 and 6. Authors simply described the observation and reluctant to give any underneath science or structure-property relationship via adding more descriptions with proper references.
2. In Figure 4. authors have shown the images of original films, these photos should be replaced with some better photos as some white portions are visible from both the images. So the authors may replace these two images.
3. Can the authors comment on the percentage of the mechanical fracture recovery, related to Figure 6, exactly or approximately, and a little bit fundamental elaboration?
4. Please re-check the English once.
Reviewer 2 Report
Shape-memory materials are important to fabricate intelligent devices in the future and can be implemented in smart switches, micropatterning, and self-healing materials. The authors in this paper reported a deformable photonic crystal that exhibits the shape memory effect upon heating. The results could help make more sophisticated intelligent materials based on the photonic crystal. Therefore, the paper is suggested to be accepted by the Materials after some revision. The questions and comments are as follow:
1. What is the time scale of the shape memory effect in the photonic crystals? Or how long does it take to achieve a full recovery after heating?
2. What is the essential unit or the most important part of the CLC-PUA composite in obtaining the shape memory effect?
3. What is the difference between LH and RH? They are almost the same except for the helical structure.
4. What is the length scale of the PUA composite film in Figures 4a and 4b? The snapshots should have a scale bar.

Reviewer 3 Report
In this paper, the authors propose and fabricate a material combining properties of stimuli responsive color and memory shape. At the first stage, a photonic crystal matrix is formed using a mixture of chiral liquid crystal and nematic reactive mesogen. Upon UV-polymerization, the unpolymerized components are eliminated and the matrix is filled with another UV-curable material which is to provide shape memory effect. The fabrication process and samples characterization are well presented. Remarkably, the authors reveal criticism to some properties of the fabricated system and instantly suggest ways to improve them, which is appealing. I believe that this is a well presented and meaningful work and would like to recommend it for publishing. My minor remarks and questions are below:
· Line 43: I suggest substituting the term “unmemorizable materials” with something more commonly used
· Could the authors specify the thickness or cell gap for optically tested films (Figures 3-4)?
· Figure 3. The authors attribute the blue shift upon polymerization to the diffusion of the CLC prepolymer. Could they describe the mechanism in more details? Why the photonic bandgap becomes narrower?
· Figure 4. Could the authors comment on the reason of further blue shift of the bandgap? Did they characterize films at step 4?
· Lines 169 and 171: check the grammar
